# Analysing Uncertainties in Offshore Wind Farm Power Output using
# Measure Correlate Predict Methodologies.
Michael Denis Mifsud[1], Tonio Sant[2], Robert Nicholas Farrugia[1]
[1]Institute for Sustainable Energy, University of Malta, Marsaxlokk, MXK1351 Malta.
[2]Department of Mechanical Engineering, University of Malta, Msida, MSD2080, Malta.
*Correspondence to:* Michael Denis Mifsud (Michael.d.mifsud.10@um.edu.mt)
Keywords: Measure Correlate Predict, Wake Model, Offshore Wind Farms, LiDAR
Abstract
*This paper investigates the uncertainties resulting from different Measure-Correlate-Predict methods*
*to project the power and energy yield from a wind farm. The analysis is based on a case study that*
*utilizes short-term data acquired from a LiDAR wind measurement system deployed at a coastal site in*
*the northern part of the island of Malta and long-term measurements from the island's international*
*airport. The wind speed at the candidate site is measured by means of a LiDAR system. The predicted*
*power output for a hypothetical offshore wind farm from the various MCP methodologies is compared*
*to the actual power output obtained directly from the input of LiDAR data to establish which MCP*
*methodology best predicts the power generated.*
*The power output from the wind farm is predicted by inputting wind speed and direction derived from*
*the different MCP methods into windPRO®[1]. The predicted power is compared to the power output*
*generated from the actual wind and direction data by using the Normalised Mean Absolute Error*
*(NMAE) and the Normalised Mean Squared Error (NMSE). This methodology will establish which*
*combination of MCP methodology and wind farm configuration will have the least prediction error.*
*The best MCP methodology which combines prediction of wind speed and wind direction, together with*
*the topology of the wind farm, is that using Multiple Linear Regression (MLR). However, the study*
*concludes that the other MCP methodologies cannot be discarded as it is always best to compare*
*different combinations of MCP methodologies for wind speed and wind direction, together with different*
*wake models and wind farm topologies.*
## 1 Introduction
The Measure-Correlate-Predict (MCP) methodology introduces uncertainty due to its inherent
statistical nature. Recent developments have seen the introduction of new computational regression
techniques such as Artificial Neural Networks (ANN) and Machine Learning, which include Decision
Trees (DT) and Support Vector Regression (SVR). In a previous study, Light Detection and Ranging
(LiDAR) data was used to compare the results of the various regression methodologies at different
LiDAR measurement heights (Mifsud, et al., 2018) with the reference site being Malta International
Airport (MIA), Luqa, and the candidate site being a coastal watch tower at Qalet Marku on the Northern
part of the island. This study uses the same wind data for the year 2016 to construct the MCP models.
However, this time the prediction is carried out on both wind speed and wind direction. Wind speed
and direction are then predicted for the period June – December 2015. This is done for the different
MCP models. The predicted wind speed and wind direction time series are then fed into a wind farm
model implemented in windPRO® Ver. 2.7 to model the overall energy yield, considering wake losses.
The power output for various wind farm configurations is obtained for each methodology. As the
LiDAR is sited on the roof of a coastal tower, at a height of 20m above mean sea level, the wind data
measured at a height of 80m would be equivalent to a wind turbine (WT) hub height of 100m above the
sea surface.
The power output in each case is compared to that obtained when the actual wind data is fed to the wind
farm model. Thus, the NMAE, the NMSE and the percentage error in the overall energy yield are

---

[1] https://www.emd.dk/windpro.

compared for the various methodologies and wind farm topologies. This is therefore a study about the uncertainties introduced by the various statistical methods, which is then further complicated by the windfarm layout. It is innovative due to the use of an MCP methodology to predict both the wind speed and the wind direction. The following literature review describes different MCP methodologies, four of which are then used in the prediction of wind speed and wind direction. The wake models are also described. This is followed by a description of the methodology used in the study, together with a description of the hypothetical wind farm used as a basis for this study. Finally, the results are presented and discussed.

## 2.  Literature Review

The first MCP methods estimated the mean long-term annual wind speed (Carta, et al., 2013). MCP methods later made use of Simple Linear Regression (SLR) (Rogers, et al., 2005) to establish a relationship between hourly wind characteristics of the candidate and the reference sites. A Multiple Linear Regression is a regression model that involves more than one regressor variable (Montgomery, et al., 2006). The regression is carried out using concurrent wind speed and wind direction data at the reference and the candidate sites. The reference site is normally the closest meteorological station e.g. airports, and the candidate site is the location chosen for the windfarm. When the model is created, hence establishing a relationship between the wind speed at both sites, the long-term wind data at the reference can be used to predict the long-term wind speed at the candidate site. More recent models established non-linear type relationships (Clive, 2004; Carta & Velazquez, 2011) by employing statistical learning (Hastie, et al., 2009). Amongst these are algorithms such as Artificial Neural Networks (ANNs) (Bilgili, et al., 2007; Monfared, et al., 2009) and the more recent Machine Learning (ML) techniques, which include Support Vector Regression (SVR) (Oztopal, 2006; Zhao, et al., 2010; Scholkopf & Smola, 2002; Alpaydin, 2010) and Decision Trees (DTs) (James, et al., 2015; Alpaydin, 2010).

A study (Carta, et al., 2013) reviewed many MCP methodologies. These included the method of ratios, first-order linear regression, higher than first-order linear methods, non-linear methods and probabilistic methods. The authors were also concerned with the uncertainties associated with MCP methodologies and argued that users of MCP methodologies have little information on which to determine the uncertainty of the methodology. One methodology to measure this uncertainty is to use the full set of data from the concurrent period to train the model and assess its quality.

Another study by Rogers compared four different MCP methodologies (Rogers, et al., 2005). These included a linear regression model, the distributions of ratios of the wind speeds at the two sites, an SVR model and another method based on the ratio of the standard deviations of the two data sets. The authors concluded that SVR gave the best results. In a different study, the same authors (Rogers, et al., 2005b) also analysed the uncertainties introduced with the use of MCP techniques. They concluded that linear regression methodologies could seriously underestimate uncertainties due to serial correlation of data. Another study shows that a proper assessment of uncertainty is critical for judging the feasibility and risk of a potential wind farm development, and the authors describe the risk of oversimplifying and assuming uncertainties (Lackner, et al., 2012).

A hybrid MCP method (Zhang, et al., 2014) which involved adding different weights depending on the distance and elevation of the candidate site to the reference sites, was applied to the input of five MCP methodologies. The methods used consisted of the Linear Regression, Variance Ratio, Weibull scale, ANNs and SVR methods. The results were assessed in terms of metrics such as the Mean Squared Error and Mean Absolute Error. Other authors (Perea, et al., 2011) evaluated three methodologies. One method included a linear regression, which was derived from the bivariate normal joint distribution and the Weibull regression method. The other method was based on conditional probability density functions applied to the joint distributions of the reference and the candidate sites. The results from these two methodologies were in turn compared to SVR. Although the conclusion was that the SVR method predicted all the parameters very accurately, the probability density function based on the Weibull distribution was better in terms of prediction accuracy.

The ability of ANNs to recognise patterns in complex data sets means that they can also be used to correlate and predict wind speed and wind direction (Zhang, et al., 2014). A neural network contains an input layer, one or more hidden layers of neurons and an output layer. A learning process updates the weights of the interconnections and biases between the neurons in the various layers. The Levenberg-Marquardt (Principe, et al., 2000) algorithm may be used for this purpose. The regression is performed by means of feedforward networks (Alpaydin, 2010) with *multilayer perceptrons* (MLP).

Another study (Velazquez, et al., 2011) utilised wind speed and direction from various reference stations. These were introduced into the input layer of an ANN. It was concluded that when wind direction was used as an angular magnitude to the input signal, the model gave better results. Estimation errors also decreased as the number of reference stations was increased. The authors concluded that ANNs are superior to other methods for predicting long-term wind data.

The use of ANNs for long-term predictions was also investigated by Bechrakis (Bechrakis, et al., 2004) using wind speed and direction measurements from just one reference station and compared these to standard MCP algorithms. This resulted in an improved prediction accuracy of 5 to 12%. Unfortunately, many models that use various reference stations use only the recorded wind speeds as input. The topologies of the ANNs used have only a single neuron in the input layer, with the output signal being the wind speed at the candidate site (Monfared, et al., 2009; Oztopal, 2006; Bilgili, et al., 2009).

Data from meteorological stations possessing long measurement periods provide a large amount of potential inputs for MCP methods. Apart from wind speed and direction, inputs can also include other climatological variables such as air temperature, relative humidity and atmospheric pressure. Hence, a multivariate MCP methodology may be utilised (Patane, et al., 2011). This technique considers all the inputs and extracts the maximum amount of information at the sites. Since some input variables may be inter-correlated, or may not provide information about the target site wind characteristics, the methodology is a two-stage process. Input variables are analysed and those that contain little or redundant information about the candidate site wind characteristics are discarded, following which, a multivariate regression is performed. It was concluded from the results of the tests made that the methodology was more accurate than standard MCP methods, with the quality of the estimation of the long-term wind resource increasing by 19%.

SVR is the adaptation of Support Vector Machines to the regression problem. This technique was developed by Vapnik (Vapnik, 1995; Vapnik, et al., 1998) to solve classification problems. SVR (Alpaydin, 2010) is popular within the renewable energy community, being a unique way to construct smooth and nonlinear regression approximations (Diaz, et al., 2017). The analysis of MCP models using SVR techniques shows that SVR is one of the techniques which best represents ML state-of-the-art (Diaz, et al., 2017). This is not only due to its prediction capability, but also to its property of universal approximation to any continuous function, and an efficient and stable algorithm that provides a unique solution to the estimation problem (Diaz, et al., 2017). Different hyperparameters were used to study the SVR methodology. Other studies describe how SVR may be adapted to wind speed prediction (Zhao, et al., 2010).

Another recent study shows the importance of DTs in improving the regression results for MCP (Diaz, et al., 2018). The study applied five different MCP techniques to mean hourly wind speed and direction, together with air density, using the data from ten weather stations in the Canary Islands. The study showed that the models using SVR and DTs provided better results than ANNs. A DT is a hierarchical data structure which implements the 'divide and conquer' rule and it may also be applied to the regression problem (Hastie, et al., 2009; Alpaydin, 2010; James, et al., 2015).

The use of LiDAR for wind resource assessment (Probst & Cardenas, 2010) shows a distinct advantage of this method over the traditional cup and wind vane measurements. This is demonstrated by studies carried out using different MCP methods such as SLR and ratio analysis. However, no analysis with ANNs, DTs or SVR is carried out. A more recent study (Mifsud, et al., 2018), which utilised the same data as this current study, analysed the accuracy of different MCP methodologies and their capability according to LiDAR measurement height. The study concluded that the MCP accuracy depended on both methodology and measurement height at the candidate site. Other studies using LiDAR at the same

measurement site were also carried out. These analysed the turbulent behaviour of the wind data
(Cordina, et al., 2017).
The issue of wake losses in a wind farm has been described by several authors and can be minimised
by optimising the layout of the wind farm (Manwell, et al., 2009). A short literature review of wake
models is now presented.
Wake models are classified into four categories (Manwell, et al., 2009) which are: Surface roughness
models (Bossanyi, et al., 1980), Semi-empirical models (Lissaman & Bates, 1977), (Vermeulen, 1980),
Eddy viscosity models (Ainslie, 1985), and Navier-Stokes solutions (Crespo & Hernandez, 1986),
(Crespo & Hernandez, 1993). A review of wind turbine wake models (Sanderse, n.d.), shows the effects
of reduced power production due to lower incident wind speed and the effect on the wind turbine rotors
due to increased turbulence. The author presents a number of reasons on why the focus on numerical
simulation is preferred to experimentation; this is mainly due to the use of Computational Fluid
Dynamics (CFD). One study presents the mathematical theory behind a simple wake model and that for
a multiple wake model (Gonzalez-Longatt, et al., 2012) while another study (Churchfield, 2013)
describes a hierarchy of wake models ranging from the empirical to large-eddy simulation (LES). Some
of the models compared include Ainslie's Model (Ainslie, 1985), Frandsen's model (Fransden, 2005),
and Jensen's Model (Jensen, 1983). The Dynamic Wake Meandering Model is another method which
is described (Larsen, et al., 2008) and also validated (Larsen, et al., 2013) in a study carried out on the
Egmond ann Zee offshore wind farm. Another study (Barthelmie, et al., 2006), compares wake model
simulations for offshore wind farms, with the wake profiles being measured by Sonic Detection and
Ranging (SoDAR). In this case, the models gave a wide range of predictions and it was not possible to
identify a model with superior projections with respect to the measurements.
In some studies, it is necessary for any wake model used to be straightforward, dependent on relatively
few wake measurements and economic in terms of the necessary computing power. Despite their
relative simplicity, these models tend to give results which are in reasonable agreement with the
available data in the case of a single wake within a small wind farm and a simple meteorological
environment. In addition, a comparison of different wake models does not suggest any particular
difference in terms of accuracy, between the sophisticated and simplified models (Manwell, et al.,
175 2009).

The use of wake models can also be illustrated by considering a semi-empirical model (Katić, et al,
1986) that is often used for wind farm output predictions. This model attempts to characterise the energy
content in the flow field whilst ignoring the details of the exact nature of the flow field, which is assumed
to consist of an expanding wake with uniform velocity deficit that decreases with distance downstream
(Manwell, et al., 2009).
The N.Ø. Jensen (Jensen, 1983) is a simple wake model based on the assumption of a wake with a linear
wake cone. The results from this model are comparable to experimental results.
Several metrics may be used to evaluate the accuracy of the models (Rogers, et al., 2005), and it is
important to employ more than one metric (Santamaria-Bonfil, et al., 2016) to perform the evaluation.
The lower the value of the metric, the better the performance of the model. In this case the Normalised
NMAE and the NMSE were used to quantify the performance of the model. The purpose of using
normalised values is to provide results which are independent of wind farm sizes (Madsen, et al., 2005).
The NMAE is suitable to describe the errors which are uniformly distributed round the mean, revealing
also the average variance between the true value and the predicted value (Hu, et al., 2013). The NMAE
applies the same weight to the individual errors. The NMSE is a measure of the extent of the dispersion
of the errors around the mean and gives a higher weight to larger errors. It assumes that the errors are
unbiased and follow a normal distribution (Santamaria-Bonfil, et al., 2016). The percentage error of the
energy yield gives an estimate of the accuracy of the model for predicting the total energy generated by
the wind farm over the period of evaluation. Due to the fact the each metric has disadvantages that can
lead to inaccurate evaluation of the results, it is not recommended to depend only on one measure
(Shcherbakov, et al., 2013).

## 3. Theoretical Background

MCP methods are based on regression techniques. Regression can be performed by using SLR. However, as mentioned above, several more powerful techniques exist amongst which are ANNs, SVR and DT. While MCP methodologies have been developed for wind speed, they cannot be directly used for predicting wind direction (Bosart & Papin, 2017). Nothing has been found in literature on MCP techniques which explicitly mentions prediction of wind direction at that candidate site. The use of wind speed vectors is a way of using a regression methodology to predict the wind direction, by breaking the wind speed vector into its respective components. MCP methodologies are normally used to predict the wind speed magnitude at the candidate site, but not the direction. Wind velocity may be negative (if one considers it as a vector) and the MCP methodology normally considers the positive value of the wind, i.e. magnitude. The methodology used creates a regression model using the wind velocity vector components to predict the wind vector components at the candidate site (Bosart & Papin, 2017).

The methodology is based upon a simple relationship between the meteorological wind direction $\theta_{met}$ and the mathematical wind direction $\theta_{math}$ such that:

$$\theta_{math} = 90 - \theta_{met} \tag{1}$$

in which the wind speed vector $\boldsymbol{V}_i$ can be broken down into its vector components such that

$$u_i = |V_i| \cos\theta_{math} = |V_i| \cos(90 - \theta_{met}) \tag{2}$$

$$v_i = |V_i| \sin\theta_{math} = |V_i| \sin(90 - \theta_{met}) \tag{3}$$

in which case the values of $u_i$ and $v_i$, which may be either positive or negative depending on the direction of the wind (the value of $\theta_{met}$), are the wind components in the North ($y$) and the East ($x$) directions (axes). The relationship is shown in Figure 1.

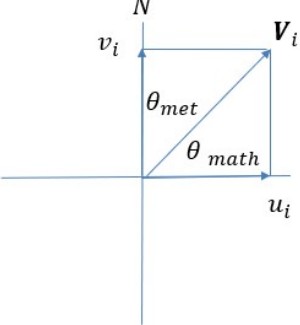

*Figure 1: Difference between the meteorological wind direction and the mathematical wind direction and the component of the wind vector.*

Also,

$$|\boldsymbol{V}_i| = \left(u_i^2 + v_i^2\right)^{\frac{1}{2}} \tag{4}$$

The regression is carried out between the respective components of the wind velocity in the $y$ and $x$ directions, hence establishing a relationship between the components at both sites. The forecasted wind direction at the candidate site is then obtained from the forecasted wind components using the relationship in Eq. (5):

$$\theta_{met_{i_p}} = 90 - tan^{-1}\frac{v_{i_p}}{u_{i_p}} \tag{5}$$

The value of the angle $\theta_{met_{i_p}}$ depends on the direction of $u_{i_p}$ and $v_{i_p}$, as shown in Figure 2

$$
\begin{array}{cc}
\begin{array}{c} u_{i_p} < 0 \\ v_{i_p} > 0 \end{array} & \begin{array}{c} u_{i_p} > 0 \\ v_{i_p} > 0 \end{array} \\
\hline
\begin{array}{c} u_{i_p} < 0 \\ v_{i_p} < 0 \end{array} & \begin{array}{c} u_{i_p} > 0 \\ v_{i_p} < 0 \end{array}
\end{array}
$$

*Figure 2: Calculating the value of $\boldsymbol{\theta}_{met_{i_p}}$ according to the value of $\boldsymbol{u}_{i_p}$ and $\boldsymbol{v}_{i_p}$.*

and in accordance with the relationships shown in Eq. (6):

$$
\begin{array}{lll}
u_{i_p} > 0 \ and \ v_{i_p} > 0 & NE \ winds & 0° < \theta_{met_{i_p}} < 90° \\
u_{i_p} > 0 \ and \ v_{i_p} < 0 & SE \ winds & 90° < \theta_{met_{i_p}} < 180° \\
u_{i_p} < 0 \ and \ v_{i_p} < 0 & SW \ winds & 180° < \theta_{met_{i_p}} < 270° \\
u_{i_p} < 0 \ and \ v_{i_p} > 0 & NW winds & 270° < \theta_{met_{i_p}} < 360°
\end{array}
\tag{6}
$$

and Eq. (7):

$$
\begin{array}{ll}
u_{i_p} = 0 \ and \ v_{i_p} > 0 \ \text{(North Wind)} & \theta_{met_{i_p}} = 0° \\
u_{i_p} = 0 \ and \ v_{i_p} < 0 \ \text{(South Wind)} & \theta_{met_{i_p}} = 180° \\
u_{i_p} > 0 \ and \ v_{i_p} = 0 \ \text{(East Wind)} & \theta_{met_{i_p}} = 90° \\
u_{i_p} < 0 \ and \ v_{i_p} = 0 \ \text{(West Wind)} & \theta_{met_{i_p}} = 270°
\end{array}
\tag{7}
$$

The results are compared by using the NMAE and the NMSE of the residuals, using the Eq (8) to Eq. (12). The residuals, $e_i$ are the errors between the predicted and the actual output power values from the windfarm,

$$
e_i = P_i - P_{act_i} \tag{8}
$$

The formula used to calculate the NMAE is shown in Eq (9), whereby the errors are normalised by dividing by the average power production over the whole period of evaluation (Madsen, et al., 2005):

$$
NMAE = \frac{\sum_{i=1}^{N} |e_i|}{\sum_{i=1}^{N} P_i} \tag{9}
$$

And the NMSE is given by:

$$
NMSE = \frac{\frac{1}{N} \sum_{i=1}^{N} (e_i)^2}{\overline{P} \cdot \overline{P_{act}}} \tag{10}
$$

where,

$$\bar{P} = \frac{1}{N} \sum_{i=1}^{N} P_i \tag{11}$$

and

$$\overline{P_{act}} = \frac{1}{N} \sum_{i=1}^{N} P_{act_i} \tag{12}$$

The percentage error in overall energy yield is given by Eq (13), where:

$$e_{eng} = \left( \frac{\sum_{i=1}^{N} P_i - \sum_{i=1}^{N} P_{act_i}}{\sum_{i=1}^{N} P_{act_i}} \right) \cdot 100\% \tag{13}$$

## 238   4. A Case Study - Site Conditions and the Modelled Offshore Windfarm

### 239   4.1 The reference and candidate sites

The reference site employed in this study is the Meteorological Office at Malta International Airport
(MIA), Luqa, and the candidate site is data collected by a ZephIR 300 LiDAR
(https://www.zxlidars.com/wind-lidars/zx-300/, n.d.) unit administered by the University of Malta's
Institute for Sustainable Energy. The unit was situated on the roof of a coastal watch tower at Qalet
Marku, situated in the Northern Part of the Island of Malta (Mifsud, et al., 2018). The relative location
of the two sites is shown in Figure 3, while Figure 4 shows a satellite image of the location of the coastal
watch tower.

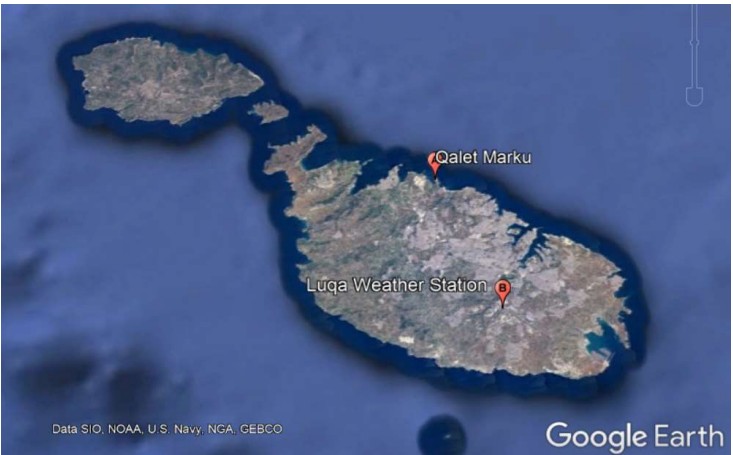


*Figure 3: Map of Malta showing relative location of the candidate and the reference sites* **(Google,**
**2019)** *(© Google Maps 2019).*

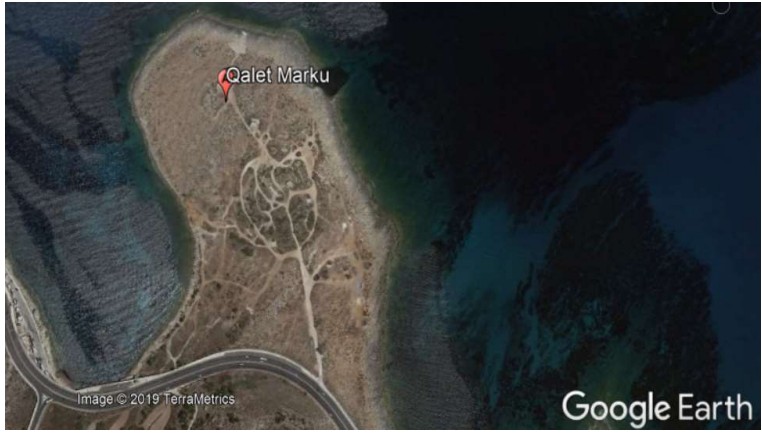


*Figure 4: Satellite imagery of the Qalet Marku coastal watch tower, located on a promontory near*
*Bahar ic-Caghaq* (Google, 2019) *(© Google Maps 2019).*
Table 1 and Table 2 show the properties of the candidate and the reference sites respectively (Cordina,
et al., 2017), (Mifsud, et al., 2018). In this case the wind data measured by the LiDAR at a height of
80m, would be equivalent to a cumulative height of 100m above sea-level, which would be the hub
height of the wind turbines in the windfarm. This is because the LiDAR is situated on the rooftop of a
coastal tower at a height of 20m above sea level,  as shown in Table 3.
*Table 1: Candidate Site parameters* (Cordina, et al., 2017).

| Station Name | Qalet Marku LiDAR Station |
|---|---|
| **LiDAR Type** | ZephIR 300 (https://www.zxlidars.com/wind-lidars/zx-300/, n.d.) |
| **Cone Angle,** **LiDAR aperture height above the tower rooftop.** | 60° 1 $m$ |
| **Measurement height, above the LiDAR aperture window, m** | 80$m$ |
| **Data** | Average hourly wind speed, wind direction, atmospheric pressure and relative humidity. |
| **Data range** | 26th June, 2015 – 31st December, 2016 |
| **Geographical Coordinates** | 35.946252°N, 14.45329°E |
| **Average tower rooftop height above surrounding ground level** | 10 $m$ |
| **Height of base of tower above sea level** | 6 $m$ |

*Table 2: Reference Site parameters (Malta International Airport).*

| Station Name | Luqa MIA Weather Station |
|---|---|
| **Measuring Instruments** | Wind – Cup and vane Digital temperature probe Digital Barometer. |
| **Data** | Average hourly wind speed, wind direction, air temperature, atmospheric pressure and relative humidity. |
| **Mast height** | 10 $m$ above ground |
| **Height of site above sea level** | 78 $m$ |
| **Geographical Coordinates** | 35.85657°N, 14.47676°E |

**4.2 The Available Wind Data**
The measurement campaign at the candidate site started on the 1st July 2015 and ended on the 31st
December 2016. Hourly wind data were available for this time period from both the reference and
candidate sites. The ideal number of data points used to create the MCP models is thus 8784, i.e. the
number of hours in 2016. Following analysis and filtration of the wind speed data at the reference site,
98% of the data was considered as suitable for the creation of the model. The data at the reference site
was all considered as suitable. Hence, the regression model was created using the concurrent 8616 wind
speed and direction values. For the year 2015, 95.6% of the data was considered valid (the measurement
campaign started on the 26th of June, 2015, hence there were 4368 hours of wind speed and direction
measurement of which 4176 were valid data points).
The MCP analysis was carried out using both wind speed and wind direction. The data from the
reference site were used as the independent data set. The models were created using the data for the
year 2016, while the reference site wind data for 2015 used to create the predicted wind speed and wind
direction as inputs to the windfarm model.

## 4.3 The Wind Farm Design in windPRO®

*Table 3: Wind Turbine Parameters used in the study* **(wind-turbine-models.com, 2019)**.

| Wind Turbine Parameter | |
|---|---|
| **Manufacturer** | RE Power (Germany) |
| **Rated Power** | 5000 $W$ |
| **Rotor orientation** | Upwind |
| **Number of blades** | 3 |
| **Rotor Diameter** | 126 $m$ |
| **Swept Area** | 12469 $m^2$ |
| **Blade Type** | LM |
| **Cut in speed** | 3.5 $ms^{-1}$ |
| **Rated Wind Speed** | 14 $ms^{-1}$ |
| **Cut out speed (for off-shore)** | 30 $ms^{-1}$ |
| **Hub-height, z** | 100 $m$ |

The hypothetical wind farm is located opposite the coastal watch tower of Qalet Marku [$14.452498°E$,
$35.945892°N$]. windPRO® 2.7 was used to render an image of the wind farm onto an image of the
LiDAR unit taken from the watch tower. This gives an indication as to the extent of the wind farm. This
is shown in Figure 5, while Figure 6 shows the satellite imagery of the wind farm, showing a 250-MW
capacity windfarm. The windfarm faces the North-West direction, which is the prevailing wind
direction.
The windfarm is made up of 50 wind turbines. There are 10 wind turbines in a row, having a cross-wind
spacing of five rotor diameters (5D). The distance between the successive rows of wind turbines, or the
downwind spacing is eight rotor diameters (8D). Thus, considering wind turbines with a rotor diameter,
$D$, of 126 $m$ (for a 5 MW Wind Turbine), the distance between the turbines in the cross-wind direction
is 630 $m$, and the distance between successive rows of wind turbines in the downwind direction is
1,008 $m$. The wind turbine selected for use in windPRO® is the RE Power 5-MW wind turbine whose
parameters are shown in Table 3.

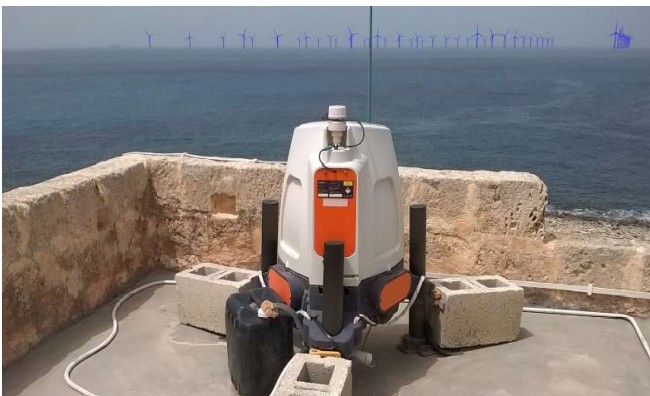

*Figure 5: View of the wind farm rendered onto an image of the area and also showing the LiDAR unit.*

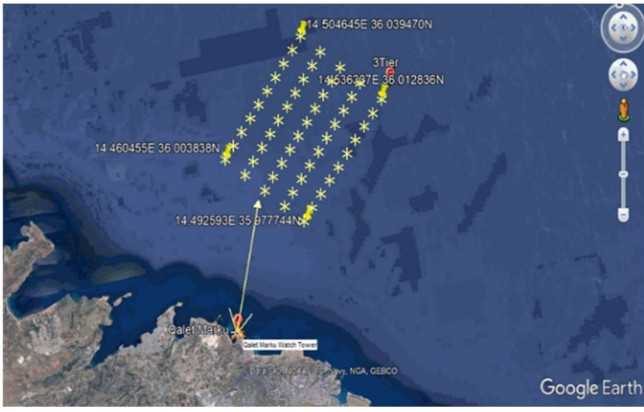


*Figure 6: Satellite imagery of the wind farm showing the location of the 50 wind turbines with respect to the coastal LiDAR*
*station* (Google, 2019) *(© Google Maps 2019).*

## 5. Methodology

Figure 7 shows the methodology applied in this paper. The study is divided into three steps as follows:
1.  STEP 1 - The various MCP methodologies are used to compute the MCP model. For wind speed,
the models are trained using wind speed and direction data at a candidate and reference site for the
year 2016. For the wind direction the input training data is the wind velocity vector component in
the North or East direction at the candidate site, and the output of the model is the respective
component at the candidate site. The models are summarised in Table 4, below. Table 4 describes
the inputs used to train the respective models, both for wind speed and wind direction. It also shows
the parameters of the models and the algorithms used to train the model, such as Least-Squares for
MLR and the Levenberg-Marquardt algorithm for ANN.
2.  STEP 2 - The 2015 wind speed and wind direction are predicted using the models computed in
Step 1. The predicted and actual wind speed and wind direction are used to compute the power
output from the wind farm. This is done by feeding the wind speed and direction data into the
windPRO® model, and,
3.  STEP 3 - compute and compare the MSE, NMAE and percentage error in the power.

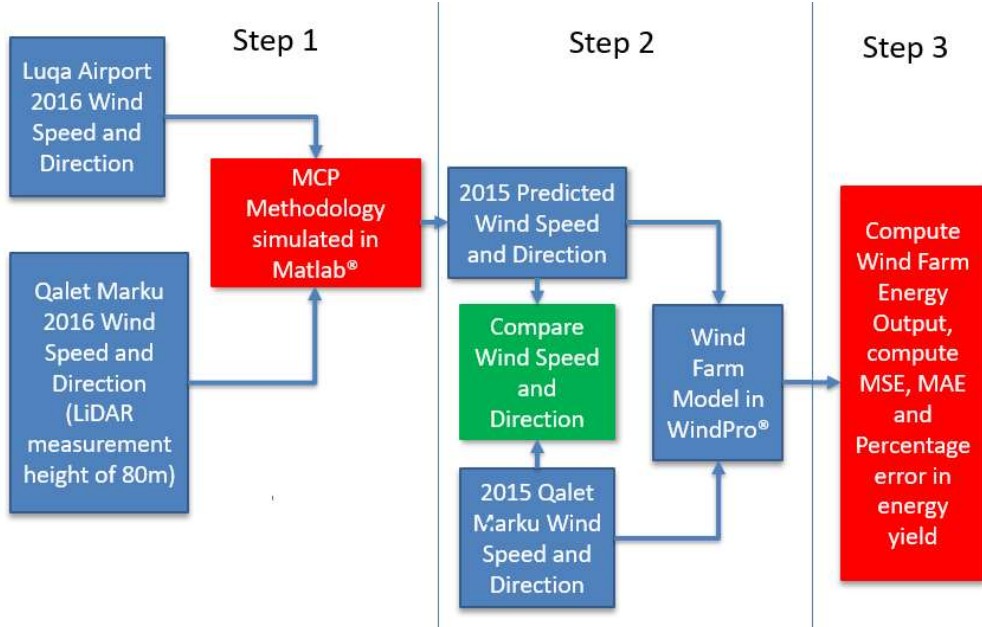


*Figure 7: Applied methodology.*
Table 4*: Description of the regression methodologies used for the Measure-Correlate-Predict Methodology.*

| MCP methodology | Wind Speed | Wind Direction |
|---|---|---|
| MLR | **Independent variables**: 2 (Wind speed magnitude, wind direction at the reference site). <br> **Dependent variables**: Wind Speed magnitude at candidate site. | **Independent variable**: Wind velocity vector in North and East direction at reference site. <br> **Dependent variable**: Wind velocity vector in North and East direction at candidate site. |
| | **Methodology**: Least Squares | |
| ANN | **Number of inputs**: 2 - Wind speed magnitude, wind direction at the reference site) <br> **Number of outputs**: 1 - Wind speed magnitude at candidate site. | **Number of inputs**: 1 - Wind velocity vector in North and East direction at reference site) <br> **Number of outputs**: 1 - Wind velocity vector in North and East direction at candidate site. |
| | **Number of layers**: 3 <br> **Number of neurons in layer**: 30,30,10 <br> **Training Methodology**: Levenberg-Marquardt Algorithm <br> **Percentage of points used for training**: 70% <br> **Percentage of points used for verification**: 15% <br> **Percentage of points used for testing**: 15% | |
| DT | **Number of inputs**: 2 - wind speed magnitude, wind direction at reference site. <br> **Number of outputs**: 1 - wind speed at candidate site. | **Number of inputs**: 1 - Wind velocity vector in North and East direction at reference site. <br> **Number of outputs**: 1 - Wind velocity vector in North and East direction at candidate site. |
| | **Number of Trees**: 200 <br> **Minimum Number of Leafs**: 5 <br> **Methodology**: Tree Bagger Ensemble | |
| SVR | **Number of inputs**: 2 - wind speed magnitude, wind direction at reference site. <br> **Number of outputs**: 1 - wind speed magnitude at candidate site. | **Number of inputs**: 1 - Wind velocity vector in North and East direction at reference site. <br> **Number of outputs**: 1 - Wind velocity vector in North and East direction at candidate site. |
| | **Methodology**: Hyperparameter optimisation, <br> **Kernel**: Gaussian <br> **Solver**: Sequential Minimal Optimisation | |

The combinations of LiDAR measurement heights and MCP methodologies are shown in Table 5.
*Table 5: Summary of combinations of methodologies, LiDAR measurement heights and amount of wind turbines used in the*
*analysis*

| | MCP Methodology | | | |
|---|---|---|---|---|
| **80m (equivalent to a 100m hub height)** | **Simple Linear Regression (SLR)** | **Artificial Neural Networks (ANN)** | **Decision Trees (DT)** | **Support Vector Regression (SVR).** |
| | Wind Speed, Wind Direction, predicted for 2015. Actual and predicted sequences fed into wind farm model, comparisons of wind farm power output made for a capacity of 250, 200, 150, 100 and 50 MW. | | | |

Regression models were created for the MCP methodologies using the reference and candidate wind
speed and direction for the year 2016. These regression models were created using SLR, ANN, DT and
SVR. A model was created for both wind speed and direction.
The wind speed and wind direction for 2015 were then predicted with the models by feeding the speed
and direction values from the reference site from the year 2015. Thus, a sequence of predicted wind
speeds and wind direction time series could be compared to the actual speed and direction measured at
the candidate site for the year 2015. The models for the wind speed and the wind direction are
independent from each other.

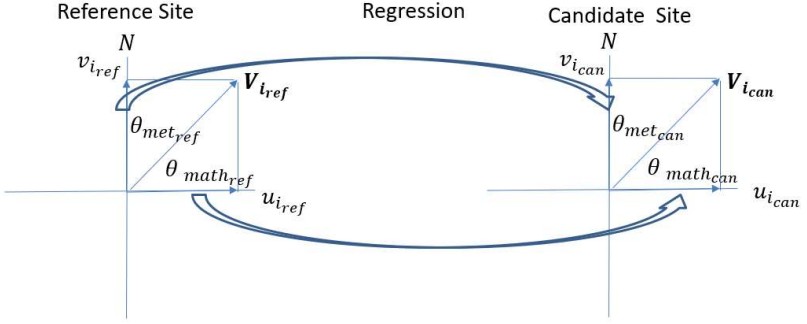

*Figure 8: Application of regression methodologies to wind direction*
In the case of wind direction, the MCP methodologies are applied as shown in Figure 8 and Figure 9.
Figure 8 shows that two regressions are carried out: one for the magnitude of the wind component in
the North direction and one for the wind component in the East direction. Thus, two models are created
using the wind speed and direction data of the reference and the candidate site for 2016. The two models
are then used to derive the predicted wind direction for 2015 at the candidate site as shown in Figure 9,
by using the wind components at the reference site for 2015 as inputs to the respective models. The
values of the wind speed in the North direction and the East direction are first predicted, and the wind
direction at the candidate site for 2015, $\theta_{met_p}$, is then derived from the mathematical relationships given
in Eq. (6) and Eq. (7).

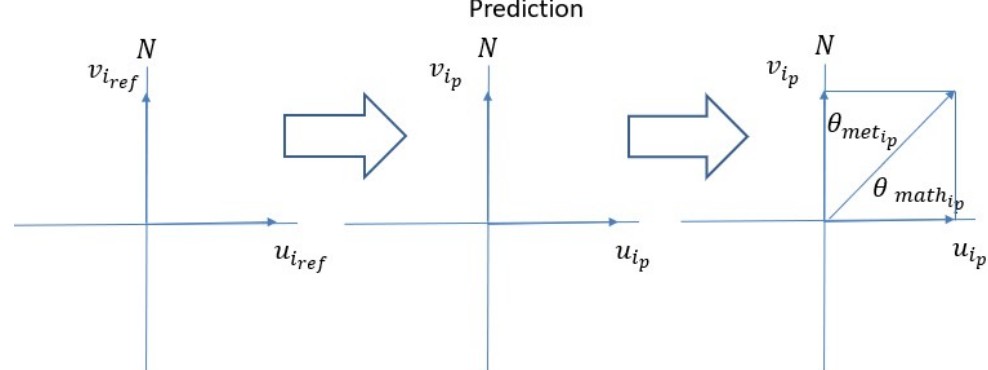

*Figure 9: Predicting the wind direction*
The sequences of wind speed and wind directions (both actual and predicted) were fed into the wind
farm model. This was done for different combinations of methodology and wind farm (250, 200, 150,
100 and 50 MW) configurations. The results were compared to determine which combination of MCP
methodology, and windfarm capacity would give the lowest prediction error. The prediction error for
the power output from the wind farm is analysed using the Mean Squared Error (MSE), the Normalised
Mean Absolute Error (NMAE) and the percentage error in the Overall Energy Yield for the period of
analysis. The results are shown in the following section.
**6. Results**
A summary of the results is shown below where sequences of data for a specific period of 2015 are
compared. These sequences are for wind speed, wind direction and power output. All MSE, NMAE and
percentage errors in the overall energy yield are then shown in the following tables.
**6.1 Wind speed and wind direction with MCP methodology.**
**6.1.1 Wind speed with MCP methodology.**
Figure 10 to Figure 13 show the wind speed from the period 23rd November to the 30th November 2015.
The particular period is chosen because of the high availability of wind. The actual wind data are
compared with that predicted by the MLR, ANN, DT and SVR methodologies. The predicted wind
values closely follow the actual wind values, for all the MCP methodologies applied.

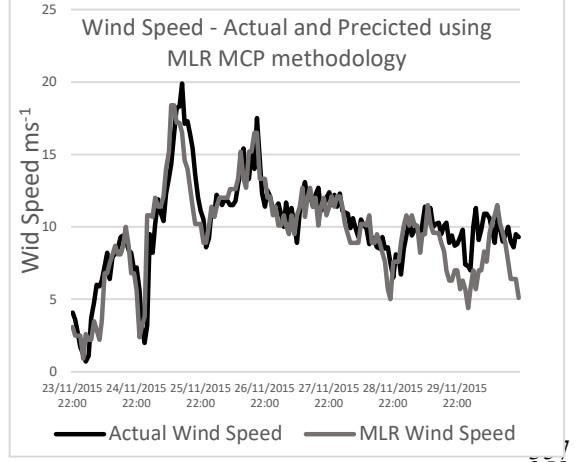

*Figure 10: Comparing actual wind speed and wind speed*
*predicted by MLR methodology with wind data for 2015.*

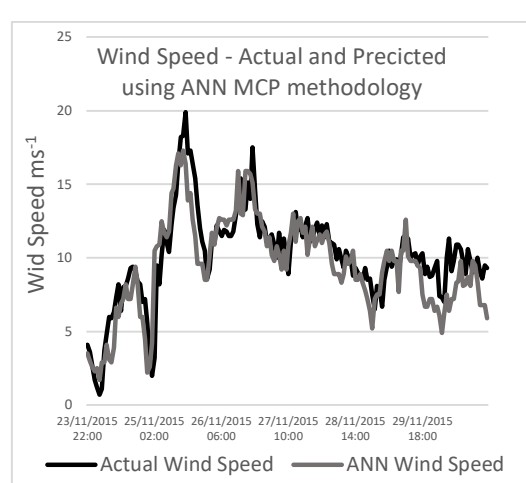

*Figure 11: Comparing actual wind speed and wind speed*
*predicted by ANN methodology with wind data for 2015.*

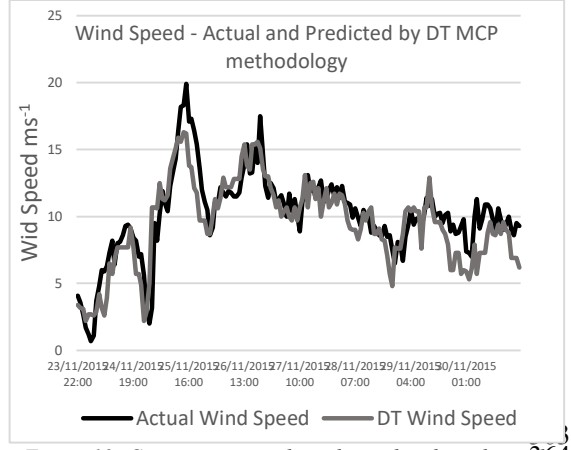

*Figure 12: Comparing actual wind speed and wind speed*
*predicted by ANN methodology with wind data for 2015.*

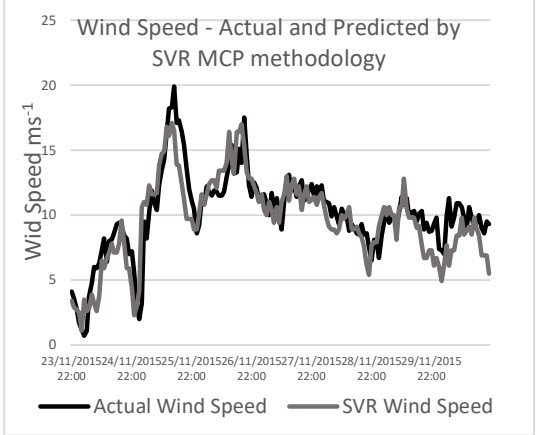

*Figure 13: Comparing actual wind speed and wind speed*
*predicted by SVR methodology with wind data for 2015.*
**6.1.2 Wind direction with MCP methodology.**
Figure 14 to Figure 17 show the wind direction from the period 23$^{rd}$ November to the 30$^{th}$ November
2015. As above, the actual wind direction at the candidate site is compared to that predicted by the
MLR, ANN, DT and SVR methodologies. Again, as in the case for wind speed, there is a similarity
between the actual and predicted wind direction values, in all cases.

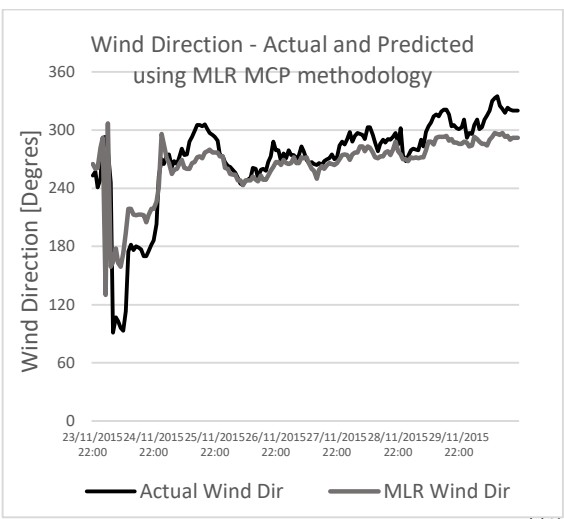

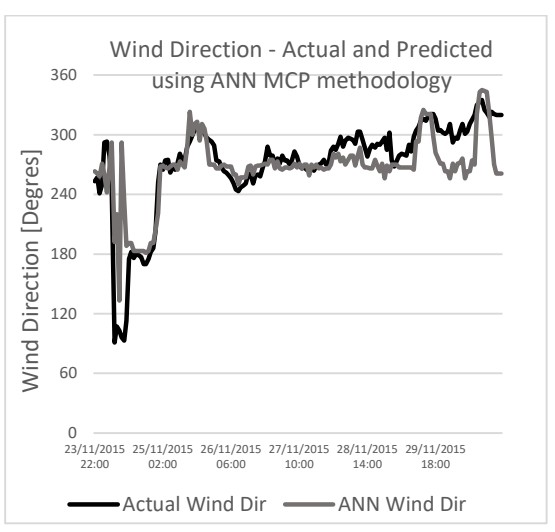

*Figure 14: Comparing actual and predicted wind direction predicted by MLR methodology, with wind data for 2015.*

*Figure 15: Comparing actual and predicted wind direction predicted by ANN methodology, with wind data for 2015.*

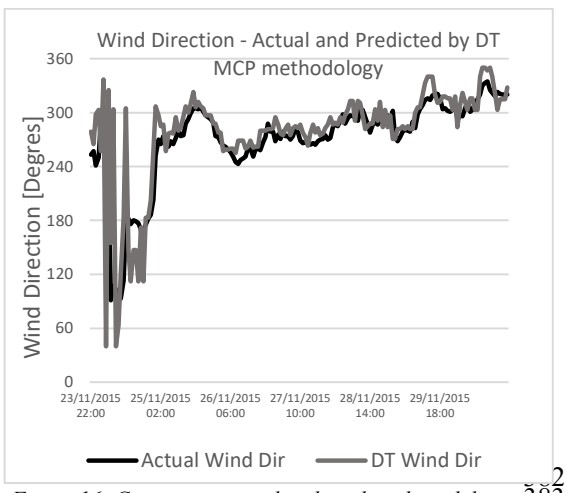

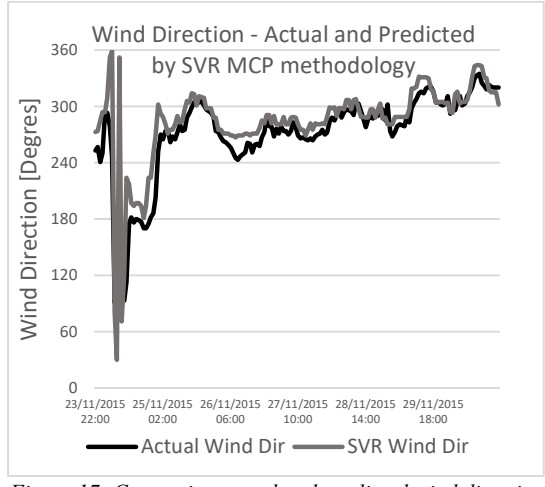

*Figure 16: Comparing actual and predicted wind direction predicted by DT methodology, with wind data for 2015.*

*Figure 17: Comparing actual and predicted wind direction predicted by SVR methodology, with wind data for 2015.*

## 6.2 Wind farm power output with MCP methodology, for a windfarm capacity of 250MW.

Figure 18 to Figure 21 compare the output power from the wind farm, which is derived from the actual wind speed and wind direction to the power output derived from the predicted wind speed and direction. This comparison is carried out for the MLR, ANN, DT and SVR methodologies. The results for a wind farm capacity of 250MW are being shown. As in the case for wind speed and direction, the predicted power output closely follows that obtained with the actual wind speed and direction.

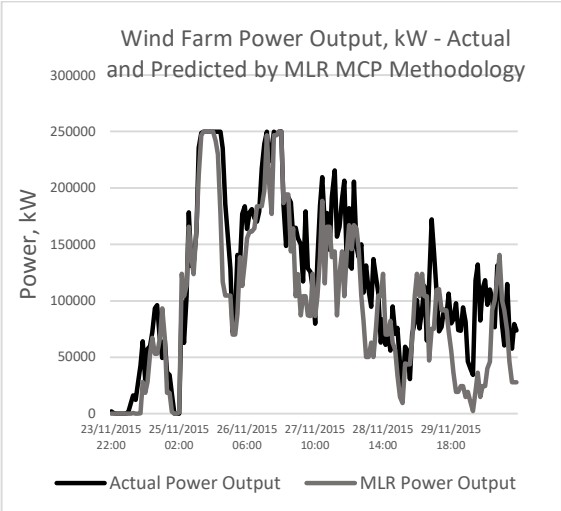

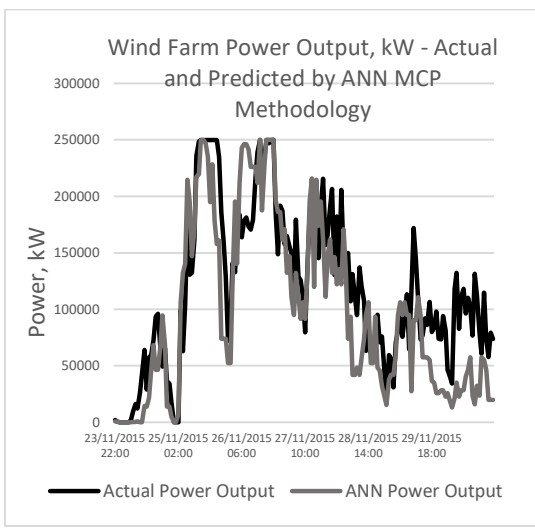

*Figure 18: Comparing actual and predicted power output from the wind farm, with wind data for 2015, actual and predicted by MLR methodology.*

*Figure 19: Comparing actual and predicted power output from the wind farm, with wind data for 2015, actual and predicted by ANN methodology.*

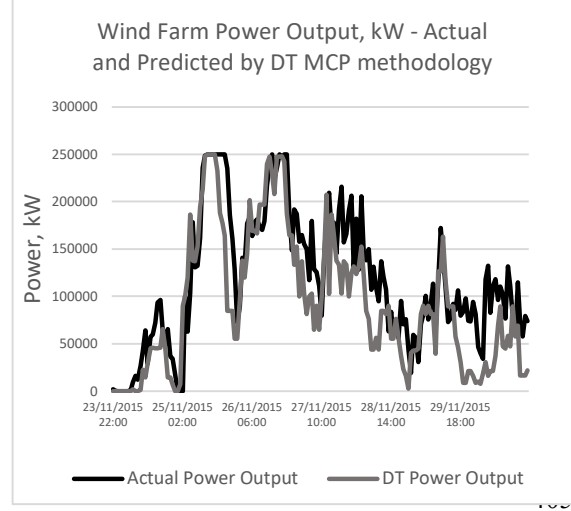

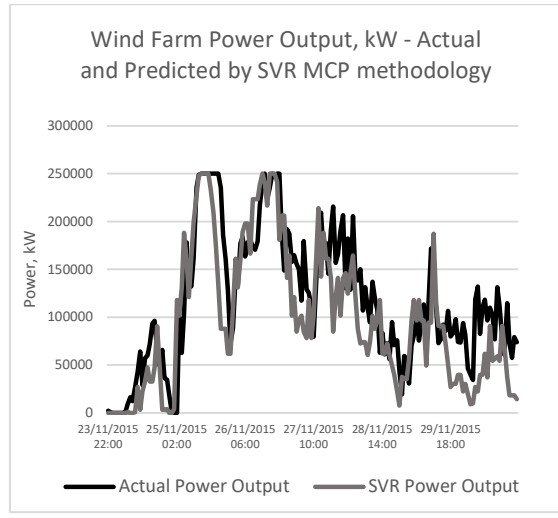

*Figure 20: Comparing actual and predicted power output from the wind farm, with wind data for 2015, actual and predicted by DT methodology.*

*Figure 21: Comparing actual and predicted power output from the wind farm, with wind data for 2015, actual and that predicted by SVR methodology.*

A Wind Data Analysis, carried out using windPRO®, is shown in the next section. The results presented are a Weibull distribution for wind speed and the wind rose. These charts are computed from the wind speed and direction which are predicted by using the MLR, ANN, DT and SVR MCP methodologies. Thus, the predicted wind speed and direction are compared with the results computed from the actual wind data.

## 6.3 The Actual Wind Data for 2015 measured by the LiDAR system.

Figure 22 shows the Wind Data Analysis report from windPRO® for the actual LiDAR data measured at the 80m level height (equivalent to a hub height of 100m). The images show the Weibull distribution for the wind speed and the wind rose. The reports are used to compare the properties of the actual wind measurements and the predicted wind speed and direction.

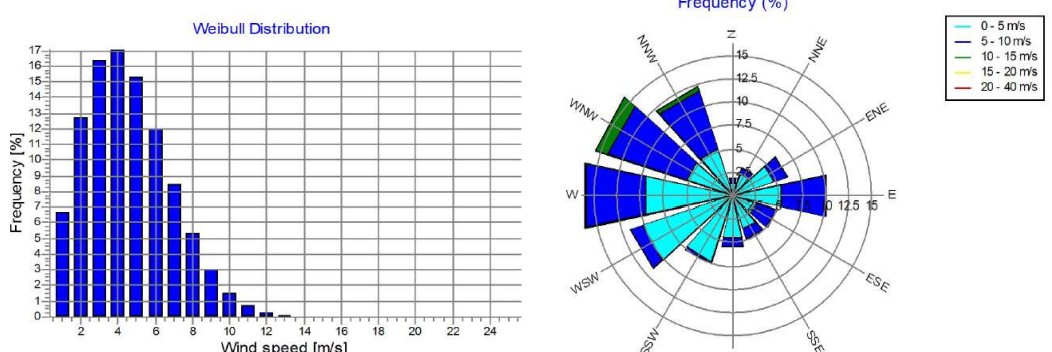

421

*Figure 22: windPRO® wind data analysis using actual wind data measured by the LiDAR equipment at a height of 100 m.*

## 6.4 Wind speed and direction predicted using the MCP methodologies.

Figure 23 to Figure 26 represent the Weibull distribution and the wind rose for the wind speed and direction predicted by the MLR, ANN, DT and SVR MCP methodologies respectively, at the hub height of 100$m$. There exists a similarity between the Weibull plots for the actual wind data and those for the predicted wind speed, for the same measurement period. While, the wind direction predicted by the ANN and DT methodologies show a higher resemblance to that of the actual wind direction than that predicted by the MLR or SVR methodologies. Hence it is expected that the ANN and the DT methodologies would yield the least error in the predicted power output from the wind farm.


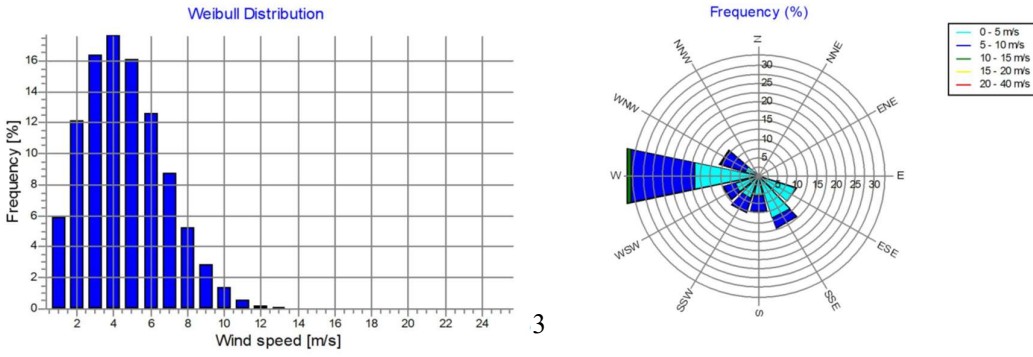


*Figure 23: windPRO® wind data analysis using wind data predicted by MCP applying MLR at a hub height of 100 m.*

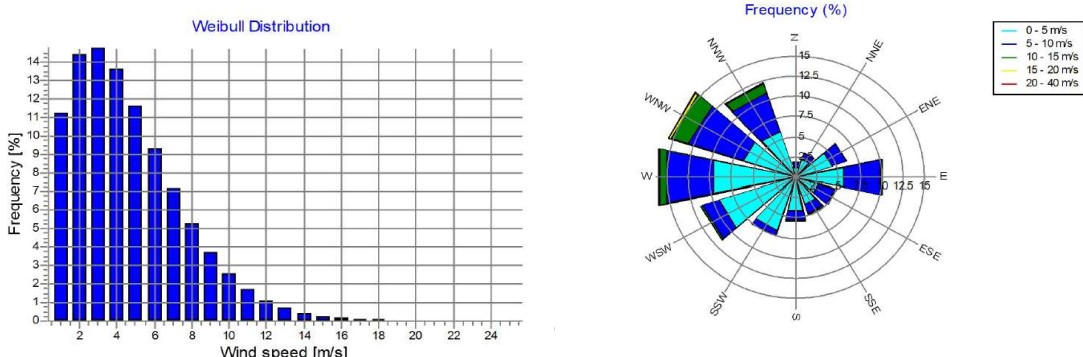


*Figure 24: windPRO® wind data analysis using wind data predicted by MCP applying ANN at a hub height of 100 m.*

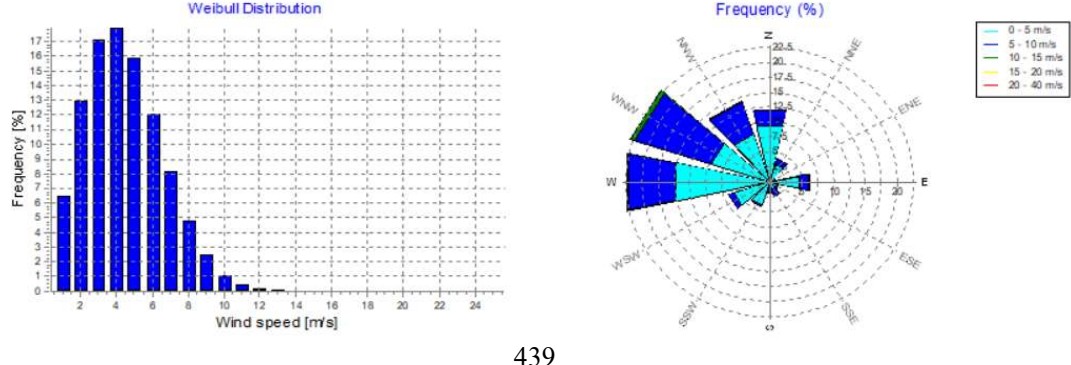

*Figure 25: windPRO® wind data analysis using wind data predicted by MCP applying DT at a hub height of 100 m*

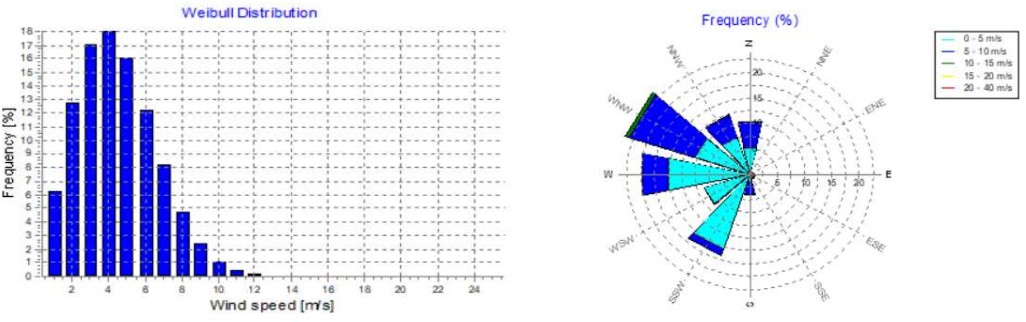

*Figure 26: windPRO® wind data analysis using wind data predicted by MCP applying SVR at a hub height of 100 m*
The results for the NMAE, the NMSE and the percentage error in the overall energy yield are
summarised in Table 6 to Table 8. The tables show that the MLR and ANN methodology have the best
performance in NMAE, NMSE and percentage error in energy yield. The results are consistent for all
wind farm capacities under consideration. ANN is better than MLR in the case of MMAE, while MLR
is slightly better than ANN in the case of the 50MW wind farm capacity. MLR is superior to ANN in
the case of NMSE for all wind farm capacities. However, the differences between the MLR and the
ANN methodologies are minimal and both methodologies show a better performance than the DT or
SVR methodologies. Especially in the case of the overall energy yield as shown in Table 8. Graphical
results are also shown in Figure 27 to Figure 29.
*Table 6: Summarised results for Normalised Mean Absolute Error (NMAE) by MCP methodology and windfarm capacity.*

| Normalised Mean Absolute Error | | | | |
|---|---|---|---|---|
| *Wind Farm Capacity* | *MLR* | *ANN* | *DT* | *SVR* |
| *250MW* | 0.505 | 0.502 | 0.572 | 0.544 |
| *200MW* | 0.502 | 0.499 | 0.565 | 0.539 |
| *150MW* | 0.492 | 0.482 | 0.545 | 0.532 |
| *100MW* | 0.484 | 0.472 | 0.537 | 0.515 |
| *50MW* | 0.510 | 0.547 | 0.573 | 0.558 |

*Table 7: Summarised results for the Normalised Mean Squared Error (NMSE) by MCP methodology and windfarm capacity.*

| Normalised Mean Squared Error |
|---|
|  |

| Wind Farm Capacity | MLR | ANN | DT | SVR |
|---|---|---|---|---|
| 250MW | 0.977 | 1.004 | 1.170 | 1.082 |
| 200MW | 0.956 | 0.979 | 1.123 | 1.052 |
| 150MW | 0.912 | 0.938 | 1.056 | 1.002 |
| 100MW | 0.834 | 0.868 | 0.960 | 0.917 |
| 50MW | 0.789 | 0.884 | 0.930 | 0.890 |

*Table 8: Summarised results for percentage error in overall energy yield by MCP methodology and windfarm capacity.*

| Percentage Error in Overall Energy Yield | | | | |
|---|---|---|---|---|
| **Wind Farm Capacity** | **MLR** | **ANN** | **DT** | **SVR** |
| 250MW | 4.63 | 4.54 | 18.83 | 9.44 |
| 200MW | 4.80 | 4.90 | 18.40 | 9.34 |
| 150MW | 4.92 | 5.40 | 17.78 | 9.23 |
| 100MW | 4.78 | 5.70 | 16.92 | 8.71 |
| 50MW | 3.65 | 7.03 | 14.73 | 8.23 |

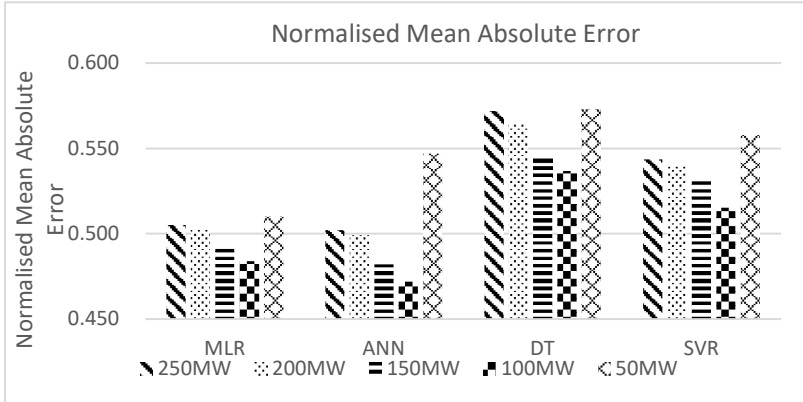


*Figure 27: Comparison of the Normalised Mean Absolute Error for the various wind farm topologies and MCP*
*methodology, for the 2015 energy output from the wind farm.*

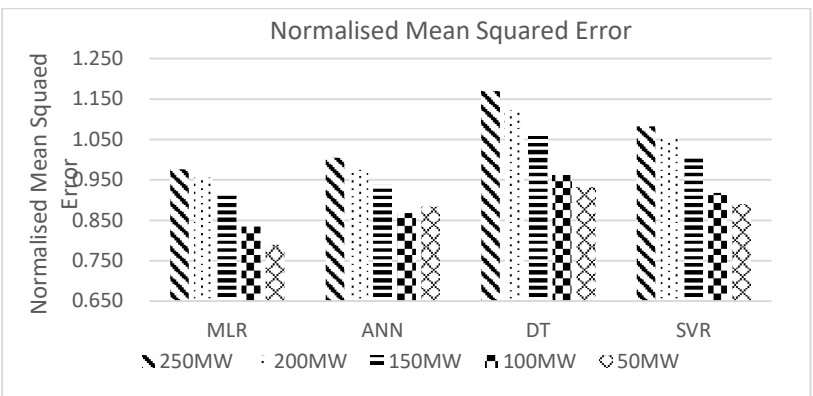


*Figure 28: Comparison of the Normalised Mean Squared Error for the various wind farm topologies and MCP*
*methodology, for the 2015 energy output from the wind farm.*

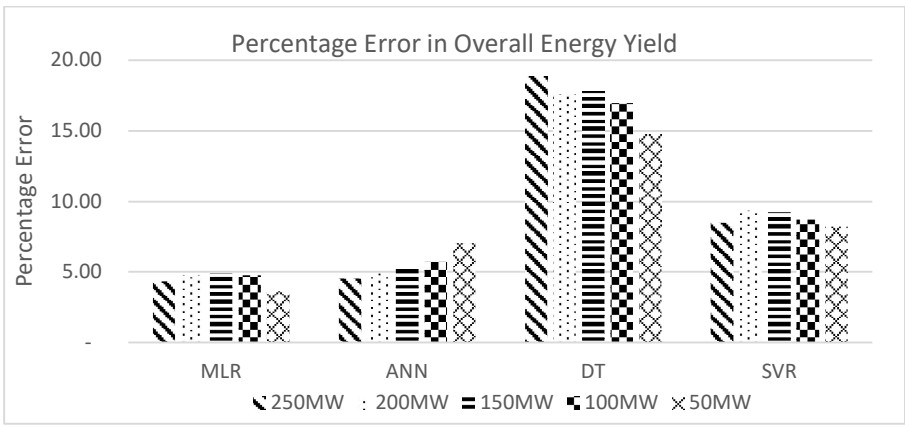


*Figure 29: Comparison of the Percentage Error in Overall Energy Yield for the various wind farm topologies and MCP*
*methodology, for the 2015 energy output from the wind farm.*
The ANN methodology also shows the best similarity to the actual wind speed and wind direction, as
seen in Figure 24. In the case of the overall energy yield, the MLR and ANN methodologies show a
significant improvement in percentage error over the DT and SVR methodologies. The ANN
methodology is only better than the MLR methodology for the 250MW windfarm capacity. The MLR
methodology has better results in the case of 200MW, 150MW, 100MW and 50MW wind farm
capacities, with the percentage error being 3.65% at a windfarm capacity of 50MW, when compared to
an error of 7.3% obtained with the ANN methodology.
Thus, the metrics show that the best methodologies for predicting the output power from the wind farm
is therefore that which uses the MLR methodology, closely followed by that which uses the ANN
methodology.

## 7. Conclusions

The above research has combined the use of MCP methodologies for wind speed and used a different
method for predicting the wind direction at a candidate site. Three of the four MCP methodologies used
are based on modern statistical learning methodologies. The data was collected from a reference site
which is the Island of Malta's international airport, while the candidate site data has been collected by
means of a LiDAR wind measurement system placed on the roof top of a coastal building.
The wind direction at the candidate site was predicted with the various MCP methodologies by breaking
down the wind velocity vector into its respective North and East direction components. The regression
analysis was then carried out on the respective components at the reference and the candidate sites. The
wind speed is predicted by using the magnitude of the wind speed at the respective sites for creating the
regression model.
The projected wind speed and direction time series were applied to a hypothetical wind farm. Thus, the
error introduced by the four MCP methods could be measured. This was done by calculating the NMAE,
the NMSE and the percentage error in wind farm's energy yield. The results show that the NMAE,
NMSE and the percentage error in energy yield depend on the MCP methodology and the windfarm
capacity, and can be used to establish an optimal MCP methodology.
In this case, the best MCP method was that which used MLR. Although other MCP methodologies gave
larger errors, they cannot be totally discarded. It is always best to compare methodologies, comparing
results by analysing residuals and errors and then choosing the best methodology on a case-by-case
basis. In this case the results from the ANN methodology gave results which are very close to the MLR
methodology, while the DT and SV methodologies gave larger errors.
Unless actual wind data is available, one cannot carry out this analysis, as the uncertainty is obtained
by comparing the energy from the windfarm with predicted and actual wind data. The above analysis
could be done because 18 months of data were available, rather than the normal 12 months, which is
usual for a wind resource assessment which uses MCP methodologies.
The above study was limited to using the same MCP methodology for both the wind speed and direction
and to the N.Ø. Jansen methodology for wake losses. The layout chosen was one that ensured a
recommended minimum distance between the wind turbines. Different combinations of MCP
methodologies for wind speed and direction can be examined.
In this case, an MCP model was created for wind speed, and two more MCP models were created for
wind speed components, which were then used to calculate the wind direction. Another possible method
is to calculate the magnitude of the wind speed from the models used to calculate the wind direction.
This was done, but, the results from the first method, were by far superior to those from the latter
method. The reason why, still needs to be investigated as part of future work, and these results are not
being presented in this paper. The advantage of having three models, also allows the possibility of using
different combinations of MCP methodologies, i.e. using MLR for wind speed and ANN for wind
direction. This was also performed for a limited number of combinations and is also the subject of
further research
Another area which warrants further study, as is trying out different windfarm topologies, or selecting
different wind turbines and different hub heights. It would also be of interest to study the application of
different wake methodologies as a possible means of decreasing the uncertainties.
**8. Author Contribution.**
Tonio Sant and Robert. N. Farrugia contributed in the preparation of the manuscript and the research
methodology.
**9. Competing Interests.**
The authors declare that they have no conflict of interest.
**10. Acknowledgements.**

522• Mr. Joseph Schiavone from the Meteorological Office at Malta International Airport, Luqa, is being
acknowledged for providing the data for the Luqa MIA Weather Station.

524• The authors would like to express their sincere gratitude to Mr. Manuel Aquilina, lab officer at the
University of Malta, for technical assistance in collecting and organising the data from the Institute for
Sustainable Energy's LiDAR system at Qalet Marku.

527• The LiDAR system was purchased through the European Regional Development Fund (ERDF 335),
part-financed by the European Union.

529• Thanks also goes to Din L'Art Ħelwa for permitting and facilitating the installation of the LiDAR unit
on the Qalet Marku Tower.

531• The windPRO® 2.7 software was funded by the project: *Setting up of Mechanical Engineering
Computer Modelling and Simulation Laboratory*, part-financed by the European Regional Development
Fund (ERDF) - Investing in Competitiveness for a Better Quality of Life, Malta 2007 – 2013.
**11. Nomenclature.**

| | |
|---|---|
| 535 ANN | Artificial Neural Network |
| 536 CFD | Computational Fluid Dynamics |
| 537 DT | Decision Trees |
| 538 LiDAR | Light Detection and Ranging |
| 539 LSE | Large Eddy Simulation |
| 540 MCP | Measure-Correlate-Predict |
| 541 MIA | Malta International Airport |
| 542 MLR | Multiple Linear Regression |
| 543 MLP | Multilayer Perceptron |

| 544 | MSE | Mean Squared Error |
|---|---|---|
| 545 | NMAE | Normalised Mean Absolute Error |
| 546 | NMSE | Normalised Mean Squared Error |
| 547 | SLR | Simple Linear Regression |
| 548 | SoDAR | Sonic Detection and Ranging |
| 549 | SVR | Support Vector Regression |
| 550 | WT | Wind Turbine |
| 551 | $V_i$ | Magnitude of wind speed in $ms^{-1}$ |
| 552 | $e_{norm_i}$ | Normalised residual |
| 553 | $e_{eng}$ | Percentage error in energy yield |
| 554 | $e_i$ | Residual, $MW$ |
| 555 556 | $u_{i_p}$ | Predicted component of wind speed vector in easterly direction at the candidate site in $ms^{-1}$ |
| 557 558 | $u_{i_{ref}}$ | Component of wind speed vector in easterly direction at the reference site in $ms^{-1}$ |
| 559 560 | $u_{i_{ref}}$ | Component of wind speed vector in easterly direction at the reference site in $ms^{-1}$ |
| 561 | $u_i$ | Component of wind speed vector in easterly direction in $ms^{-1}$ |
| 562 563 | $v_{i_{can}}$ | Component of wind speed vector in northerly direction at the candidate site in $ms^{-1}$ |
| 564 565 | $v_{i_p}$ | Predicted component of wind speed vector in northerly direction at the candidate site in $ms^{-1}$ |
| 566 567 | $v_{i_{ref}}$ | Component of wind speed vector in northerly direction at the reference site in $ms^{-1}$ |
| 568 | $v_i$ | Component of wind speed vector in northerly direction in $ms^{-1}$ |
| 569 | $z_0$ | surface roughness |
| 570 | $\mathbf{V}_i$ | Wind speed vector (speed in $ms^{-1}$, wind direction in $deg$) |
| 571 | $\theta_{math_{i_p}}$ | Predicted mathematical wind direction at the candidate site in $deg$ |
| 572 | $\theta_{met_{i_p}}$ | Predicted meteorological wind direction at the reference site in $deg$ |
| 573 | $\theta_{met_{can}}$ | Meteorological wind direction at the candidate site in $deg$ |
| 574 | $\theta_{met_{ref}}$ | Meteorological wind direction at the reference site in $deg$ |
| 575 | $\theta_{math}$ | Mathematical wind direction |
| 576 | $\theta_{met}$ | Meteorological wind direction |
| 577 | $D$ | Wind turbine diameter, $m$ |
| 578 | $N$ | Number of data points |
| 579 | $P$ | Predicted power output from wind farm, $MW$ |
| 580 | $P_{act}$ | Actual power output from windfarm, $MW$ |
| 581 | | |

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
