# Peer review of "Measure Correlate Predict Methodologies."

_Wind Energy Science, 2019_

## Referee Comment (RC1) · Anonymous Referee #1 · 12 Dec 2019

This article is focused in the field of prediction of wind conditions and wind farm power. In particular, a comprehensive literature review on the related methodologies, such as Measure Correlate Predict methods, LiDAR and Wake models is performed. Very useful contribution. I have only minor comments that aims to help the reader understand better your work:

1. Table 1 and 2 do not give enough information. For example, 'Data' in Table 1 needs to list the specific parameters instead of just highlighting the data interval.

2. On line 179, 'While MCP methodologies have been developed for wind speed, they cannot be directly used for predicting wind direction.'. Could you explain this?

3. On line 243, you said 'SSTEP 1 – The various MCP methodologies are used to compute the MCP model. This is done using wind speed and direction data at a candidate and reference site for the year 2016'. However, the paper lacks the description of the modelling. For the regression model, how many inputs are you use? Are these MCP models one-step-ahead prediction model? What are the other settings in these models? For example, how many hidden layers are there in the ANN and what type of the hidden neurons are selected. If the modelling information is provided, it will be clearer and easier to understand.

4. You mentioned that the models were created using the data for the year 2016. Have you checked that the amount of data is enough to create a satisfactory MCP model?

---

## Referee Comment (RC2) · Anonymous Referee #2 · 6 Jan 2020

[referee-annotated manuscript omitted]

---

## Author Comment (AC1) · 3 Feb 2020

The authors would like to thank the reviewer for the valuable comments provided. The comments are answered below and the changes to the paper will be highlighted in yellow, while the changes which common to all reviewers are highlighted in light blue.

1. **Table 1 and 2 do not give enough information. For example, 'Data' in Table 1 needs to list the specific parameters instead of just highlighting the data interval.**

Tables 1 and 2 have been modified as follows:
- Table 1:
  - LiDAR instrumentation type
  - Type of data measured by the LiDAR
- Table 2:
  - Met station instrumentation.

A reference to the LiDAR instrumentation has been included in line 239:

(https://www.zxlidars.com/wind-lidars/zx-300/, n.d.)

The tables 1 and 2 are shown below with the modifications to the tables being highlighted in yellow.

*Table 1: Candidate Site parameters (Cordina, et al., 2017).*

| Station Name | Qalet Marku LiDAR Station |
|---|---|
| **LiDAR Type** | ZephIR 300 (https://www.zxlidars.com/wind-lidars/zx-300/, n.d.) |
| **Cone Angle,** **LiDAR aperture height above the tower rooftop.** | 60° 1 $m$ |
| **Measurement height, above the LiDAR aperture window, m** | 80$m$ |
| **Data** | Average hourly wind speed, wind direction, atmospheric pressure and relative humidity. |
| **Data range** | 1$^{st}$ July, 2015 – 31$^{st}$ December, 2016 |
| **Geographical Coordinates** | 35.946252°$N$, 14.45329°$E$ |
| **Average tower rooftop height above surrounding ground level** | 10 $m$ |
| **Height of base of tower above sea level** | 6 $m$ |

*Table 2: Reference Site parameters (Malta International Airport).*

| Station Name | Luqa MIA Weather Station |
|---|---|
| **Measuring Instruments** | Wind – Cup and vane Digital temperature probe Digital Barometer. |
| **Data** | Average hourly wind speed, wind direction, air temperature, atmospheric pressure and relative humidity. |
| **Mast height** | 10 $m$ above ground |

| | Height of site above sea level | $78\ m$ | |
|---|---|---|---|
| | **Geographical Coordinates** | $35.85657°N, 14.47676°E$ | |

| | |
|---|---|
| **2.** | **On line 179, 'While MCP methodologies have been developed for wind speed, they cannot be used directly for predicting wind direction.'. Could you explain this?** |

Nothing has been found in literature on Measurement-Corelate-Predict techniques which explicitly mentions prediction of wind direction at the candidate site. A reference on the use of vectors was found in a presentation by Bosart and Papin (Bosart & Papin, 2017), which showed a way of using a regression methodology to predict the wind direction, by breaking the wind speed vector into its respective components. MCP methodologies are normally used to predict the wind speed magnitude at the candidate site, not the direction. The methodology used creates a regression model using the wind velocity vector components to predict the wind vector components at the candidate site, hence deriving the wind direction. Bosart and Papin's method is adapted, in this paper, to MCP methodologies.

**This clarification will be included in the paper at line 197 as follows.**

"While MCP methodologies have been developed for wind speed, they cannot be directly used for predicting wind direction. Nothing has been found in literature on Measurement-Corelate-Predict techniques which explicitly mentions prediction of wind direction at that candidate site. The use of wind speed vectors is a way of using a regression methodology to predict the wind direction, by breaking the wind speed vector into its respective components. MCP methodologies are normally used to predict the wind speed magnitude at the candidate site, but not the direction. Wind velocity may be negative (if one considers it as a vector) and the MCP methodology normally considers the positive value of the wind, i.e. magnitude. The methodology used creates a regression model using the wind velocity vector components to predict the wind vector components at the candidate site (Bosart & Papin, 2017)."

| | |
|---|---|
| **3.** | **On line 243, you said 'SSTEP 1 – the various MCP methodologies are used to compute the MCP model. This is done using wind speed and direction data at a candidate and reference site for the year 2016'. However, the paper lacks the description of the modelling. For the regression model, how many inputs are you use? Are these MCP models one-step ahead prediction model? What are the other settings in these models? For example, how many hidden layers are there in the ANN and what type of hidden neurons are selected. If the modelling information is provided, it will be clearer and easier to understand.** |

The MCP methodologies used in this paper are described by (Mifsud, et al., 2018). The figures reproduced below are from the reference and show a description of the ANN model used for the regression between the candidate and the reference site.

[Figure]

Fig. 7. Model used for Artificial Neural Network for regression between the reference site and the candidate site.

**Table 6**
Characteristics of the ANN used to compute the regression between the wind speed and direction at the reference site and the wind speed at the candidate site.

| | | |
|---|---|---|
| Number of inputs | 2 | Wind speed in $ms^{-1}$ and wind direction in degrees, at the reference site (Luqa Weather Station). |
| Number of outputs | 1 | Wind speed in $ms^{-1}$ at the candidate site (LiDAR) |
| Number of layers | 3 | |
| Number of neurons in layer | 30,30,10 | |
| Training methodology. | Levenberg-Marquardt algorithm | |
| Percentage of points used for training. | 70% | |
| Percentage of points used for verification of model | 15% | |
| Percentage of points used for testing of model | 15% | |

The Multiple Linear Regression (MLR), Artificial Neural Network (ANN), Decision Trees (DT) and Support Vector Regression (SVR) models used for the prediction of wind speed, use wind speed (magnitude) and wind direction (in degrees) as input, and the wind speed at the candidate site as the target data to train the model. The models are created using 2016 wind data and 2015 wind data at the reference site is fed into the model to predict the 2015 wind speed at the candidate site.

[Figure]

Fig. 6. Steps for constructing the regression model using 2016 data and predicting the 2015 wind speed.

The reference paper describes the MLR, Decision Tree (DT) and the Support Vector Regression (SVR) models. The data and methodologies are the same for this paper. The paper

also describes the mathematical theory of the MCP methodologies and how they are applied to predict the wind at the candidate site.

MCP models are not one step ahead prediction models.

The same model structure is used for the prediction of wind direction. The input training data in this case is the vector component in the North or East Direction at the candidate site and the output of the model is the respective component at the candidate site (for 2016). The reference site data for 2015 is then run through the model to predict the north and east components of the wind. The wind direction is then derived.

**Table 4 (below) will be introduced as a description of the models used in the MCP, and a description of the contents of the table will be included in line 293, as follows:**

1. STEP 1 - The various MCP methodologies are used to compute the MCP model. For wind speed, the models are trained using wind speed and direction data at a candidate and reference site for the year 2016. For the wind direction the input training data is the wind velocity vector component in the North or East direction at the candidate site, and the output of the model is the respective component at the candidate site. The models are summarised in Table 4, below. Table 4 describes the inputs used to train the respective models, both for wind speed and wind direction. It also shows the parameters of the models and the respective algorithms used to train the model, such as Least-Squares for MLR and the Levenberg-Marquardt algorithm for ANN.

*Table 4: Description of the regression methodologies used for the Measure-Correlate-Predict Method*

| MCP methodology | Wind Speed | Wind Direction |
|---|---|---|
| SLR | **Independent variable**: Wind speed magnitude at reference site. **Dependent variable**: Wind Speed magnitude at candidate site. | **Independent variable**: Wind velocity vector in North and East direction at reference site. **Dependent variable**: Wind velocity vector in North and East direction at candidate site. |
|  | **Methodology**: Least Squares | |
| ANN | **Number of inputs**: 2 - wind speed magnitude, wind direction at the reference site. **Number of outputs**: 1 - wind speed magnitude at candidate site. | **Number of inputs**: 1 - Wind velocity vector in North and East direction at reference site. **Number of outputs**: 1 - Wind velocity vector in North and East direction at candidate site. |
|  | **Number of layers**: 3 **Number of neurons in layer**: 30,30,10 **Training Methodology**: Levenberg-Marquardt Algorithm **Percentage of points used for training**: 70% **Percentage of points used for verification**: 15% **Percentage of points used for testing**: 15% | |
| DT | **Number of inputs**: 2 - wind speed magnitude, wind direction at reference site. **Number of outputs**: 1 - wind speed at candidate site. | **Number of inputs**: 1 - Wind velocity vector in North and East direction at reference site. **Number of outputs**: 1 - Wind velocity vector in North and East direction at candidate site. |

| | | |
|---|---|---|
| | **Number of Trees**: 200
**Minimum Number of Leafs**: 5
**Methodology**: Tree Bagger Ensemble | |
| SVR | **Number of inputs**: 2 - Wind speed magnitude, wind direction at reference site.
**Number of outputs**: 1 - Wind speed magnitude at candidate site. | **Number of inputs**: 1 - Wind velocity vector in North and East direction at reference site.
**Number of outputs**: 1 - Wind velocity vector in North and East direction at candidate site. |
| | **Methodology**: Hyperparameter optimisation,
**Kernel**: Gaussian
**Solver**: Sequential Minimal Optimisation | |

4. **You mentioned that the models were created using the data for the year 2016. Have your checked that the amount of data is enough to create a satisfactory MCP model?**

1. MCP are normally carried out using hourly wind data measured over the period of a year. This means that for 2016 there are 8784 data points, which is considered adequate and within the scope of the MCP methodology.

**Lines 58 and line 261 have been modified accordingly:**

**Line 58:**
The regression is carried out using concurrent wind speed and wind direction data at the reference and the candidate sites. The reference site is normally the closest meteorological station e.g. airports, and the candidate site is the location chosen for the windfarm. When the model is created, hence establishing a relationship between the wind speed at both sites, the long-term wind data at the reference can be used to predict the long-term wind speed at the candidate site.

**Line 261:**
The ideal number of data points used to create the MCP models is thus 8784, the number of hours in 2016. Following analysis and filtration of the wind speed data at the reference site, 98% of the data was considered as suitable for the creation of the model. The data at the reference site was all considered as suitable. Hence, the regression model was created using the concurrent 8616 wind speed and direction values. For the year 2015, 95.6% of the data was considered valid (the measurement campaign started on the 26th of June, 2015, hence there were 4368 hours of wind speed and direction measurement of which 4176 were valid data points).

References

Bosart, L. & Papin, P., 2017. *www.atmos.albany.edu.* [Online] Available at: www.atmos.albany.edu/.../2017/pptx/ATM305_Statistics_16Nov17.pptx [Accessed 3 March 2019].

Cordina, C., Farrugia, R. & Sant, T., 2017. *Wind Profiling using LiDAR at a Costal Location on the Mediterranean Island of Malta.* s.l., s.n.

https://www.zxlidars.com/wind-lidars/zx-300/, n.d. [Online] [Accessed 19 January 2020].

Mifsud, M., Sant, T. & Farrugia, R., 2018. A comparison of Measure-Correlate-Pedict Methodologies using Lidar as a Candidate Site Measuement Device for the Mediterranean Island of Malta. *Renewable Energy,* Issue 127, pp. 947-959.

---

## Author Comment (AC2) · 3 Feb 2020

The authors would like to thank the reviewer for the valuable comments provided. The comments are answered below and the changes to the paper will be highlighted in green, while the changes which are answers common to all reviewers are highlighted in light blue.

**Line 40: I don't understand this statement. Why are they equivalent?**

Line 41 reworded as follows for clarity:

As the LiDAR is sited on the roof of a coastal tower, at a height of 20m above the mean sea level,, the 80m measurement height would be equivalent to an offshore wind turbine (WT) hub height of 100m above the sea surface.

**Line 160: meandering?**

Line 162:

Dynamic Wake Meandering Model

**Line: 216: again, confusing**

**Clarified in line 250 as follows:**

In this case the wind data measured by the LiDAR at a height of 80m, would be equivalent to a cumulative height of 100m above sea-level, which would be the hub height of the wind turbines in the windfarm. This is because the LiDAR is situated on the rooftop of a coastal tower at a height of 20m above sea level, as shown in Error! Reference source not found.Table 3.

**Line 223: what was the % of data that could be used?**

**Inserted the below in line 261:**

Following analysis and filtration of the wind speed data at the reference site, 98% of the data was considered as suitable for the creation of the model. The data at the reference site was all considered as suitable. Hence, the regression model was created using the concurrent 8616 wind speed and direction values. For the year 2015, 95.6% of the data was considered valid (the measurement campaign started on the 26th of June, 2015, hence there were 4368 hours of wind speed and direction measurement of which 4176 were valid data points).

**Line 236: in a row**

**Line 280 changed as follows for a better clarification:**

The windfarm is made up of 50 wind turbines. There are 10 wind turbines in a row, having a cross-wind spacing of five rotor diameters (5D). The distance between the successive rows of wind turbines, or the downwind spacing is eight rotor diameters (8D).

**Line 278: So, if I understand correctly, an MCP model is made for wind speed. Then for wind direction, two more MCP models are created for the wind speed components, which are then used to calculate the wind direction. These latter two models could of course also be used to calculate the magnitude of the wind speed and compared to the first MCP model. Was that done? Is there an advantage of one approach versus the other? Please comment.**

You understand correctly. This was done, but the results obtained with the first method (3 MCP models), were, by far superior to the second method (2 MCP models used to calculate the magnitude and direction of the wind). The reason why still remains to be investigated, and these results are not being presented in this paper. The scope of having three models, also possibly allows analysis of different combinations of MCP methodologies, i.e. using MLR for wind speed and ANN for wind direction. This was done for a limited number of combinations and is the subject of further research.

The results presented are those using 3 MCP models of the same type, and a comparison is thus made using four regression methodologies.

**This paper is modified to reflect this in line 502, as part of the conclusion:**

In this case, an MCP model was created for wind speed, and two more MCP models were created for wind speed components, which were then used to calculate the wind direction. Another possible method is to calculate the magnitude of the wind speed from the models used to calculate the wind direction. This was done, but, the results from the first method, were by far superior to those from the latter method. The reason why, still needs to be investigated as part of future work, and these results are not being presented in this paper. The advantage of having three models, also allows the possibility of using different combinations of MCP methodologies, i.e. using MLR for wind speed and ANN for wind direction. This was also performed for a limited number of combinations and is also the subject of further research.

**Line 393: More info needed on how these values were calculated. i.e. what formulas, etc.**

**Also, would tables 5 and 6 be more informative if they were normalized by the wind farm capacity, or average power output of the farm?**

The residual values are being changed to normalised values, based on the average of the residuals. There the following paragraph is being introduced to show the formulas used to calculate the metrics. The formulas used to derive these metrics are inserted as follows:

**Line 18:**

*The predicted power is compared to the power output generated from the actual wind and direction data by using the Normalised Mean Absolute Error (NMAE) and the Normalised Mean Squared Error (NMSE).*

Line 44:

Thus, the NMAE, the NMSE and the percentage error in the overall energy yield are compared for the various methodologies and wind farm topologies.

Line 228

The results are compared by using the NMAE and the NMSE of the residuals, using the Eq (8) to Eq. (12). The residuals, $e_i$ are the errors between the predicted and actual output power values from the windfarm,

$$e_i = P_i - P_{act_i} \qquad (8)$$

The formula used to calculate the NMAE is shown in Eq. (9), whereby the errors are normalised by dividing by the average power production over the whole period of evaluation (Madsen, et al., 2005):

$$NMAE = \frac{\sum_{i=1}^{N} |e_i|}{\sum_{i=1}^{N} P_i} \qquad (9)$$

And the Normalised Mean Square Error (NMSE) is given by:

$$NMSE = \frac{\frac{1}{N} \sum_{i=1}^{N} (e_i)^2}{\bar{P} \cdot \overline{P_{act}}} \qquad (10)$$

where,

$$\bar{P} = \frac{1}{N} \sum_{i=1}^{N} P_i \qquad (11)$$

and

$$\overline{P_{act}} = \frac{1}{N}\sum_{i=1}^{N} P_{act_i} \tag{12}$$

The percentage error in overall energy yield is given by Eq (13), where:

$$e_{eng} = \left(\frac{\sum_{i=1}^{N} P_i - \sum_{i=1}^{N} P_{act_i}}{\sum_{i=1}^{N} P_{act_i}}\right) \cdot 100\% \tag{13}$$

The nomenclature is modified accordingly:

Line 553: $e_i$              Residual, *MW*
Line 552: $e_{eng}$          Percentage error in energy yield
Line 550: NMSE       Normalised Mean Squared Error
Line 545: NMAE       Normalised Mean Absolute Error
Line 577: $N$             Number of data points
Line 578: $P$             Predicted power output from wind farm, *MW*
Line 579: $P_{act}$          Actual power output from windfarm, *MW*

Thus tables 6 and 7 are modified as follows:

*Table 16: Summarised results for Normalised Mean Absolute Error by MCP methodology and windfarm capacity.*

**Normalised Mean Absolute Error**

| Wind Farm Capacity | MLR | ANN | DT | SVR |
|---|---|---|---|---|
| 250MW | 0.505 | 0.502 | 0.572 | 0.544 |
| 200MW | 0.502 | 0.499 | 0.565 | 0.539 |
| 150MW | 0.492 | 0.482 | 0.545 | 0.532 |
| 100MW | 0.484 | 0.472 | 0.537 | 0.515 |
| 50MW | 0.510 | 0.547 | 0.573 | 0.558 |

*Table 27: Summarised results for the Normalised Mean Squared Error (NMSE) of the normalised residuals by MCP methodology and windfarm capacity.*

**Normalised Mean Squared Error**

| Wind Farm Capacity | MLR | ANN | DT | SVR |
|---|---|---|---|---|
| 250MW | 0.977 | 1.004 | 1.170 | 0.082 |
| 200MW | 0.956 | 0.979 | 1.123 | 1.052 |
| 150MW | 0.912 | 0.938 | 1.056 | 1.002 |
| 100MW | 0.834 | 0.868 | 0.960 | 0.917 |
| 50MW | 0.789 | 0.884 | 0.930 | 0.890 |

**Line 425: I am having a hard time interpreting the results. Fundamentally, I don't see how we should distinguish between the three metrics used - MAE, MSE, and percentage error. What**

**do they each represent, and why are they not essentially equivalent? A reader needs more information of how to interpret the results and why the three metrics are each important.**

The equations for the NMAE, NMSE and percentage error are now included in lines 227 to 235. Results are now normalised.

Many references describe the use of multiple metrics to judge the quality of regression statistics (Rogers, et al., 2005), and it is important to employ more than one metric (Santamaria-Bonfil, et al., 2016). The lower the value, the better the performance of the model. Hence, the model having the lowest NMAE and NMSE, have the best performance. NMAE and NMSE are used to quantify the performance of the models. While NMAE is suitable for describing uniformly distributed errors. It also reveals any average variance between the forecast value and the true value (Hu, et al., 2013). The NMAE gives the same weight to the errors, while the NMSE gives a larger weight to the larger errors, and avoids using the absolute value.

The NMSE assumes that the errors are unbiased and follow a normal distribution. The percentage error in energy yield gives an estimate of the accuracy of the model in the long-term, as it is the difference of the sum of the total predicted energy generated over the period of evaluation, expressed as a percentage of the actual energy.

Hence an evaluation of the three metrics is necessary to evaluate the quality of the models.

**This is inserted in the paper in line 182 as follows:**

Several metrics may be used to evaluate the accuracy of the models (Rogers, et al., 2005), and it is important to employ more than one metric (Santamaria-Bonfil, et al., 2016) to perform the evaluation. The lower the value of the metric, the better the performance of the model. In this case the NMSE and the NMSE were used to quantify the performance of the model. The NMAE is suitable to describe the errors which are uniformly distributed round the mean, revealing also the average variance between the true value and the predicted value (Hu, et al., 2013). The NMAE applies the same weight to the individual errors. The NMSE is a measure of the extent of the dispersion of the errors around the mean and gives a higher weight to larger errors. It assumes that the errors are unbiased and follow a normal distribution (Santamaria-Bonfil, et al., 2016). The percentage error of the energy yield gives an estimate of the accuracy of the model for predicting the total energy generated by the wind farm over the period of evaluation. Due to the fact the each metric has disadvantages that can lead to inaccurate evaluation of the results it is not recommended to depend only on one measure (Shcherbakov, et al., 2013)..

**References**

Hu, J., Wang, J. & Zeng, G., 2013. A Hybrid Forecasting Approach Applied to Wind Speed Time Series. *Renewable Energy,* Volume 60, pp. 185 - 194.

Rogers, A., Rogers, J. & Manwell, J., 2005. Uncertainties in Results of Measure-Correlate-Predict Analyses. *American Wind Energy Association,* May.

Santamaria-Bonfil, G., Reyes-Ballestros, A. & Gershenson, C., 2016. Wind Speed Forecasting for Wind Farms: A Method Based on Support Vector Regression. *Renewable Energy,* Volume 85, pp. 790-809.

Shcherbakov, M. et al., 2013. A survey of Forecast Error Measures. *World Applied Sciences Journal,* Volume 24, pp. 171 - 176.